# Peer review of "Expectations and Prospects of Young Adult Caregivers Regarding the Support of Professionals: A Qualitative Focus Group Study"

_ijerph, 2020, doi:10.3390/ijerph17124299_

Round 1
Reviewer 1 Report
Abstract: The wording of the results section of the abstract is unclear. If you divided them into processes and actual support, might make more sense to say you had two overarching themes with five sub-themes overall. This seems more consistent with the way the results section is presented.
Introduction:
Would be strengthened if the first para focused on young care givers and the impact it has on them. The stats on children living with chronically ill family member distract from the overall purpose.
Is the aim to understand the support young care givers need to manage their own health and wellbeing, or support to provide care, or both? Could be clearer.
Methods:
It states that the focus is on young adult caregivers enrolled in study. But does this have any relationship to the topic under investigation, or is it just the most suitable/convenient setting for recruitment. This needs to be clearer if the study itself does not focus on the role of education in the caregiving experience. If there is an interest in how eduction level/type influences the role of caregivers, this is not clear in the stated aim or purpose of the study.
Would expect to see a clearer statement of the questions/topics expored during the focus groups. This will influence interpretation of the results.
Results:
Overall the results are clearly and logically structured. I have made some minor comments on the manuscript.
It would improve the flow of the results description if the participant number was provided with the quote. For example, instead of saying 'as stated by participant 13 in focus group 4' in text you could say one participant stated...
‘And I actually need to have a connection with a care professional. We need to click, I think. If I click with them, I can tell them anything. But if I don't click, then, um, I just rush through things a bit [Participant 13, FG4).’
Discussion:
First few paragraphs appear to mostly restate the results.
Although the study defines professionals as both health and education professioanals, their roles in a caregiver's life will be different and this does not come through in the results or discussion section. It may be necessary to be clearer about the implications for the two sectors in the discussion section.
See other comments on the manuscript.
Conclusion:
Needs a stronger statements about the significance and implications of the findings.

Reviewer 2 Report
This was a fascinating paper and one that addresses an issue rarely thought about. My main concern was that I did not feel the gender imbalance was fully acknowledged or explored. 25 participants, only one of whom is male warranted much more discussion and debate than was present here. To say that it may be young carers wish to become nurses as they have developed an orientation towards helping others is not an adequate response. I would expect to see debate around the traditional gender role of females taking on caring roles and how that is replicated (or not) through the research. I would also want there to be more acknowledgement of the bias inherent in the research of which students volunteered to be part of the study and a more nuanced debate as to why that might be so - again simply stating that young carers are likely go into caring professions is superficial and simplistic. It could be that professional programmes have an additional duty of care to applicants in terms of their caring experiences and how they might shape their perspective.
